# Causal Analytical Framework for Time-Varying Dynamics of Hallucinations in LLMs

## Abstract

Hallucinations in Large Language Models (LLMs) have emerged as a critical bottleneck for LLMs' application, causing misjudgments, amplifying bias, polluting information and eroding trust.

Prior studies on faithfulness hallucinations mainly focus on the contextual disconnection and the question-answer mismatch hallucinations. The former involves the internal inconsistency of the generated sequence, which is manifested as a logical contradiction or semantic break with the previous output. The latter relates to the the external inconsistency between the model and the user's intent, causing the answer to deviate from the question. However, current methods lack causal awareness and overlook dynamic evolution of hallucinations. To address these drawbacks, we introduce a *Causal Analytical Framework for Time-Varying Dynamics of Hallucinations* in LLMs. Specifically, we first clarify the unbiased causal effects of the prefix and the problem on the current generated sequence. Next, we propose a *Time-Varying Causal Hallucination Index System* to measure the contextual disconnection hallucination and the question-answer mismatch hallucination. Overall, our work has the following highlights:

(1) *Causal tracing.* We achieve the identification of causal pathways and the interpretable tracing of the root of hallucinations. (2) *Precise and dynamic quantification.* This framework describes the spatio-temporal dimensions of autoregressive hallucination generation, providing quantitative support for analysis and risk monitoring. (3) *Reference-free.* Our indexes effectively monitor hallucinations without standard answers, enabling unified measurement in no-ground-truth settings.

## 1 Introduction

Hallucinations in Large Language Models (LLMs) have emerged as a critical bottleneck for LLMs' application, causing misjudgments, amplifying bias, polluting information and eroding trust (Huang et al., 2025). The LLM faithfulness hallucinations can be categorized into two types: the *contextual disconnection hallucination* and the *question-answer (QA) mismatch hallucination*. The former involves the internal inconsistency of the generated sequence, which is manifested as a logical contradiction or semantic break with the previous output. The latter relates to the the external inconsistency between the model and the user's input, reflecting the model's misinterpretation of the user's intent, causing the answer to deviate from the question.

For the contextual disconnection hallucination, prior works mainly strengthen token-level correlations, such as enhancing attention weights (Chuang et al., 2024), fine-tuning long-context attention for distant dependencies (Liu et al., 2025), and using sliding windows/segmentation for span-level coherence (Xiao et al., 2023). These designs enhance representation alignment and similarity signals. For the QA mismatch hallucination, typical strategies target topical overlap or surface compatibility: retrieval augmentation injects external knowledge to raise relevance and then uses static correlation-based scores during decoding (Lewis et al., 2020), whereas semantic matching estimates QA agreement in embedding/entailment space via snapshot similarity metrics that do not track stepwise evolution (Manakul et al., 2023).

However, these methods suffer from two fundamental limitations. Firstly, they *lack causal awareness*: they only rely on correlational signals without explaining why generation deviates from logical

consistency. Secondly, they *overlook the dynamic evolution* of error propagation, where small-probability biases in autoregressive decoding can cascade into severe mismatches. Together, these shortcomings give rise to two key challenges for existing solutions. On the one hand, they struggle to uncover the mechanisms behind semantic breakdown. On the other hand, they fail to explain the butterfly-effect amplification, which often serves as the underlying cause of hallucination.

To address these drawbacks, we propose the *Causal Analytical Framework for Time-Varying Dynamics of Hallucinations* in LLMs. This framework identifies hallucinations by focusing on causality rather than correlation, leveraging causal analysis to distinguish true hallucinatory patterns from spurious correlational signals. Furthermore, it breaks down the autoregressive generation process of LLM by time step, thereby achieving a characterization of the generation process's dynamic evolution.

However, implementing this framework poses two challenges. Firstly, superficial correlations and deep causal relations in LLM outputs are intertwined, making it difficult to extract true causal drivers from non-causal information. Secondly, a mathematical model of sequence generation is essential to evaluate such hallucinations. The model should not only accurately represent the temporal dependencies in the generation process, but also have the ability to make hallucinations measurable and traceable. Existing frameworks fail to describe the coupling between generated content and the prefix in the temporal dimension, limiting the ability to accurately determine real causal associations.

To tackle the first challenge, we employ the Structural Causal Model (SCM) (Pearl, 2010) for attribution to identify sources of the hallucinations. We firstly model the sequence generation process with the SCM linking the input problem, prefix, and current generation. Next, we perform deconfounded interventions on confounders impacting sequence generation to reveal unbiased causal effects between the factors rather than mere correlations.

To combat the second challenge, we introduce the *Time-varying Causal Hallucination Index System* that render hallucination measurable and traceable. The proposed system includes the *Contextual Disconnection Hallucination Index* (CDHI) and the *Question-Answer Mismatch Hallucination Index* (QAMI). CDHI quantifies the coupling between current generation and prefix at each timestep. QAMI assesses the genuine causal linkage between input and output semantics.

To the best of our knowledge, we are the first to quantitatively detect LLM hallucinations with a focus on causality and time-varying dynamics. Functionally, CDHI targets the contextual disconnection hallucination, successfully characterizing the relationship between current and prefix outputs. QAMI reflects the QA mismatch hallucination, exhibiting a consistent discriminative trend with common evaluation metrics in cross-model experiments.Our contributions are as follows:

(1) *Causal tracing.* We are the first to model and attribute LLM hallucinations from a causal perspective. Based on which we achieve the identification of causal pathways and the interpretable tracing of the root of hallucinations. As a causality-based method, our approach enhances systematic understanding and theoretical depth of understanding LLM hallucinations.

(2) *Precise and dynamic quantification.* Our proposed system can quantitatively express and dynamically capture hallucinations, which focuses on describing the spatio-temporal dimension of the autoregressive generation of hallucinations, providing time-varying and quantitative support for hallucination analyzing and risk monitoring.

(3) *Reference-free.* Our framework does not depend on the standard answer. They shift evaluation from offline reference comparison to online process monitoring. This method minimizes subjectivity and costs linked to reference construction and allows unified measurement of hallucination types in no-ground-truth settings.

## 2 STRUCTURAL CAUSAL MODELING

We model sequence generation as an autoregressive time-step process. First, the question $Q$ is tokenized and decomposed into model-recognizable basic units, denoted as $Q_{\text{token}}$. At time $t = 1$, the decoder generates the first token $\varepsilon_1$ conditioned on $Q_{\text{token}}$, yielding the initial answer $A_1 = \varepsilon_1$. At $t = 2$, using the generated prefix $A_1 = \varepsilon_1$ and decoding under $Q_{\text{token}}$, the model produces $\varepsilon_2$. Then, the model concatenates $\varepsilon_2$ with $\varepsilon_1$ to form $A_2 = \varepsilon_1\varepsilon_2$. Similarly, at $t = 3$, the model uses $A_2$

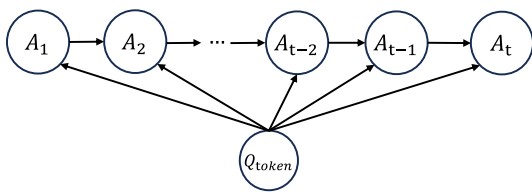

Figure 1: Causal structure of sequence generation in LLMs.

and $Q_{\text{token}}$ to generate $\varepsilon_3$, yielding $A_3 = \varepsilon_1\varepsilon_2\varepsilon_3$. This process continues until an end-of-sequence token (`<EOS>`) is produced or a maximum length is reached.

Specifically, when generating $\varepsilon_t$, the model computes the conditional distribution $P(\varepsilon_t \mid Q_{\text{token}}, A_{t-1})$ based on the already generated prefix $A_{t-1}$ and the question $Q_{\text{token}}$, and then selects the most likely token for coherent sequence generation. In summary, the probability of each token depends on the previously generated prefix and the original question, namely $\varepsilon_t$ is influenced by $Q_{\text{token}}$ and $A_{t-1} = \varepsilon_1\varepsilon_2\ldots\varepsilon_{t-1}$. Thus, with the deterministic update $A_t = \text{concat}(A_{t-1}, \varepsilon_t)$, the sequence $A_t$ is influenced by $Q_{\text{token}}$ and the preceding sequence $A_{t-1}$.

Based on the above analysis, we create a structural causal graph for sequence generation in LLMs, as shown in Figure 1. The structural causal graph presents two types of causal paths. The first is $A_{t-2} \to A_{t-1} \to A_t$, which shows that the sequence generated at each time step is influenced by the previous one. The second is $Q_{\text{token}} \to \{A_{t-2}, A_{t-1}, A_t\}$, which shows that $Q_{\text{token}}$ affects the generation at each time step.

As shown in Figure 1, for the current sequence $A_t$, the causal paths can be divided into two categories. The first is direct causal path like $A_{t-1} \to A_t$ and $Q_{\text{token}} \to A_t$. While the second is causal path with confounding factors affecting both the prefix $A_{t-1}$ and the current sequence $A_t$, which results in biased and incorrect estimation of the causal effect from $A_{t-1}$ to $A_t$. Specifically, $Q_{\text{token}}$ acts as a confounder between $A_{t-1}$ and $A_t$, leading to the observed influence of $A_{t-1}$ on $A_t$ partially comes from the influence of $Q_{\text{token}}$ through $A_{t-1}$ on $A_t$. Thus, the observed influence of $A_{t-1}$ on $A_t$ is exaggerated.

To eliminate amplified effects, exclusion experiments can be considered. However, these experiments are often time-consuming, costly, and complex to execute, making them difficult to scale in practice. Thus, we adopt an approach based on the SCM (Pearl, 2010). This method avoids expensive exclusion trials by employing causal graph modeling and intervention analysis based on the backdoor criterion, which is a crucial aspect of causal inference that helps block confounding paths. Specifically, this approach enables the efficient and low-cost elimination of the confounding influence of $Q_{\text{token}}$ on $A_t$, thereby revealing the true causal effect between relevant factors.

To express the causal relationships in Figure 1, we present the equations of the SCM:

$$Q_{\text{token}} = f(\varepsilon_Q), \quad A_1 = f(Q_{\text{token}}, \varepsilon_1), \quad A_2 = f(Q_{\text{token}}, A_1, \varepsilon_2), \ldots, A_t = f(Q_{\text{token}}, A_{t-1}, \varepsilon_t).$$

To further quantify the causal dynamics encapsulated by these structural equations and make the causal effect computable by eliminating the do-operator, we derive the following theorem regarding the unbiased causal effect.

**Theorem 1.** *The unbiased causal effect of $A_{t-1}$ on $A_t$ which is obtained by intervening on $Q_{token}$, is given by:*

$$P(A_t \mid do(A_{t-1} = a_{t-1})) = \sum_{Q_{token}} P(\varepsilon_t \mid Q_{token}, A_{t-1} = a_{t-1}) \cdot P(Q_{token}) \tag{1}$$

*Proof.* $P\big(A_t \mid \text{do}(A_{t-1} = a_{t-1})\big)$ denotes the probability of the generated sequence variable $A_t$ when applying the do-operation to set $A_{t-1} = a_{t-1}$. The do-operator is central to intervention analysis in the SCM, assigning fixed values to variables and cutting off the influence paths of confounders (Pearl, 2009). We remove the equation where $A_{t-1}$ is the dependent variable in the structural equations and replace it with the fixed value $a_{t-1}$. In the original structural equation, $A_{t-1}$ is determined by the previous sequence $A_{t-2}$ and the encoded question $Q_{\text{token}}$, namely,

$$A_{t-1} = f\big(Q_{\text{token}}, A_{t-2}, \varepsilon_{t-1}\big).$$

To implement the intervention, remove this equation so that $A_{t-1}$ is no longer affected by $Q_{\text{token}}$ and $A_{t-2}$, and is directly assigned the fixed value $a_{t-1}$. Next, substitute $A_{t-1} = a_{t-1}$ into all subsequent equations. For example, consider the generation equation of $A_t$,

$$A_t = f(Q_{\text{token}}, A_{t-1}, \varepsilon_t).$$

After the intervention $\mathrm{do}(A_{t-1} = a_{t-1})$, it becomes:

$$A_t = f(Q_{\text{token}}, a_{t-1}, \varepsilon_t).$$

As a result, the generation of $A_t$ only depends on $Q_{\text{token}}$, the fixed value $a_{t-1}$, and random noise $\varepsilon_t$, eliminating the confounding influence of $Q_{\text{token}}$ via $A_{t-1}$. This operation cuts off the correlation between $A_{t-1}$ and $Q_{\text{token}}$, blocking the backdoor path $A_{t-1} \leftarrow Q_{\text{token}} \rightarrow A_t$. Thus, Theorem 1's expression removes the confounding effect of $Q_{\text{token}}$, allowing $P(A_t \mid \mathrm{do}(A_{t-1} = a_{t-1}))$ to reflect the true causal effect of $A_{t-1}$ on $A_t$, rather than a mere statistical correlation. This procedure distinguishes the "spurious association" from the "true causal effect" in large language model generation, providing a foundation for unbiased effect estimation. $\qquad\square$

**Theorem 2.** *When there is no confounding factor exists between $Q_{token}$ and $A_t$, the causal view reduces to the correlational perspective, namely:*

$$P(A_t = a_t \mid do(Q_{token} = q)) = P(A_t = a_t \mid Q_{token} = q).$$

We construct the SCM that captures the causal dependencies between the input question and the generated sequences at each time step. Unlike traditional correlation-based methods, our approach utilizes the SCM to carry out intervention analysis. This removes confounding effects and distinguishes spurious associations from true causal effects, offering a deeper perspective for hallucination detection. We then obtain unbiased causal effects of the prefix and the question on the current generated sequence. Based on this, we create an index system to detect two main types of hallucinations.

## 3 TIME-VARYING CAUSAL HALLUCINATION INDEX SYSTEM

As we discussed in the previous section, the contextual disconnection hallucination and the QA mismatch hallucination affect the reliability and usability of generated content, reducing the quality of the answers generated by LLMs.

Structural causal modeling reveals that these hallucinations stem from confusion between the causal effects and the statistical correlations. The contextual disconnection hallucination disrupts the temporal causal chain, while the QA mismatch hallucination diminishes the causal influence of the question on the answer. We propose the *Time-varying Causal Hallucination Index System* to transform LLM hallucination assessment from empirical description to causal explanation.

### 3.1 THE CONTEXTUAL DISCONNECTION HALLUCINATION INDEX

To detect internal inconsistencies in the generated context of LLMs, we define the *Contextual Disconnection Hallucination Index* (CDHI). It quantifies the hallucination caused by insufficient semantic coherence between the current generation and its prefix.

**Definition 1.** *We define the Contextual Disconnection Hallucination Index (CDHI) at time step $t$ as:*

$$CDHI(A_t = a_t \mid A_{t-1} = a_{t-1}) = \log_2 \frac{P(A_t = a_t)}{P(A_t = a_t \mid do(A_{t-1} = a_{t-1}))}. \tag{2}$$

Furthermore, we have the following theory on the computability of CDHI:

**Theorem 3.** *CDHI can be computed as:*

$$CDHI(A_t = a_t \mid A_{t-1}) = \log_2 \frac{\sum\limits_{Q_{token}=q} P(A_t = a_t \mid Q_{token} = q) \cdot P(Q_{token})}{\sum\limits_{Q_{token}=q} P(\varepsilon_t \mid Q_{token} = q, A_{t-1} = a_{t-1}) \cdot P(Q_{token})}. \tag{3}$$

*Proof.* Specifically, $\text{CDHI}(A_t = a_t \mid A_{t-1} = a_{t-1})$ measures the log-ratio of the probabilities of generation with and without prefix $A_{t-1}$ guidance. For the denominator, according to Pearl (2010), we intervene on $A_{t-1} = a_{t-1}$ to block the backdoor path from the question variable $Q_{\text{token}}$, thereby eliminating confounding effects:

$$\text{CDHI}(A_t = a_t \mid A_{t-1}) = \log_2 \frac{P(A_t = a_t)}{\sum\limits_{Q_{\text{token}}=q} P(A_t = a_t \mid Q_{\text{token}} = q, A_{t-1} = a_{t-1}) \cdot P(Q_{\text{token}})}. \quad (4)$$

For the numerator, by applying the law of total probability, we have:

$$\text{CDHI}(A_t = a_t \mid A_{t-1}) = \log_2 \frac{\sum\limits_{Q_{\text{token}}=q} P(A_t = a_t \mid Q_{\text{token}} = q) \cdot P(Q_{\text{token}})}{\sum\limits_{Q_{\text{token}}=q} P(A_t = a_t \mid Q_{\text{token}} = q, A_{t-1} = a_{t-1}) \cdot P(Q_{\text{token}})}. \quad (5)$$

When the current sequence $A_t$ differs from the prefix $A_{t-1}$ only in the new token $\varepsilon_t$, the probability difference arises solely from $\varepsilon_t$. Thus, given $Q = q$ and $A_{t-1} = a_{t-1}$, the probability of $A_t = a_t$ can be expressed as the probability of $\varepsilon_t$, leading to equation (3). $\qquad\square$

### 3.2 THE QUESTION-ANSWER MISMATCH HALLUCINATION INDEX

To characterize hallucinations from inconsistencies between the question and the generated answer, we define the *Question-Answer Mismatch Hallucination Index* (QAMI), which quantifies the discrepancy between the probabilities of an answer with and without the question prompt. This helps identify whether the illusion stems from a lack of correlation between the question and the answer, namely "the answer is off-topic."

**Definition 2.** *Suppose the generated answer to the question $Q = q$ is $A_n = a_n$. We define the Question-Answer Mismatch Hallucination Index (QAMI) as:*

$$QAMI(A_t = a_n \mid Q = q) = \log_2 \frac{P(A_n = a_n)}{P(A_n = a_n \mid do(Q = q))}. \quad (6)$$

Furthermore, we have the following theory on the computability of QAMI:

**Theorem 4.** *The calculation for QAMI is as follows:*

$$QAMI(A_t = a_n \mid Q = q) = \log_2 \frac{P(\varepsilon_1 = a_1) \cdot \prod_{t=1}^{n} P(\varepsilon_t = a_t \mid A_{t-1} = a_{t-1})}{P(\varepsilon_1 = a_1 \mid Q = q) \cdot \prod_{t=1}^{n} P(\varepsilon_t = a_t \mid Q = q, A_{t-1} = a_{t-1})}. \quad (7)$$

*Proof.* According to the backdoor criterion(Pearl, 2010), we apply intervention to block the causal effect of the question. We have:

$$QAMI(A_t = a_n \mid Q = q) = \log_2 \frac{P(A_n = a_n)}{P(A_n = a_n \mid Q = q)}. \quad (8)$$

We further decompose $P(A_n = a_n \mid Q = q)$ based on the chain ruleMurphy (2012). As the model generates answers autoregressively, the generation of each token $\varepsilon_t$ depends on the question $Q = q$ and the prefix $A_{t-1}$. We fix the prefix $A_{t-1}$ to the partial answer $a_{t-1}$. Thus, the full answer generation probability can be expressed as:

$$P(A_n = a_n \mid Q = q) = P(\varepsilon_1 = a_1 \mid Q = q) \cdot \prod_{t=1}^{n} P(\varepsilon_t = a_t \mid Q = q, A_{t-1} = a_{t-1}). \quad (9)$$

Similarly, we have:

$$P(A_n = a_n) = P(\varepsilon_1 = a_1) \cdot \prod_{t=1}^{n} P(\varepsilon_t = a_t \mid A_{t-1} = a_{t-1}). \quad (10)$$

Substituting equation (9) and equation (10) into equation (8), we can derive equation (7). $\qquad\square$

Specifically, $QAMI(A_t = a_n \mid Q = q)$ represents the log-ratio of the probability of generating the answer without question guidance versus with question guidance. A smaller value indicates strong question guidance. Therefore, the risk of QA mismatch hallucination is low. Conversely, a larger QAMI suggests limited guidance and implying a higher risk of QA mismatch hallucination.

## 4 EXPERIMENTAL EVALUATION

We focus on news summarization with the CNN/DailyMail dataset (See et al., 2017) to assess models' hallucination tendencies. Models evaluated include the T5 family (Raffel et al., 2020), Qwen2.5-Instruct (Team, 2024), and Mistral-v0.1 (Jiang et al., 2023). Experiments are conducted on a single NVIDIA A100 GPU (80 GB).

### 4.1 VALIDATION ON HALLUCINATION INDEXES

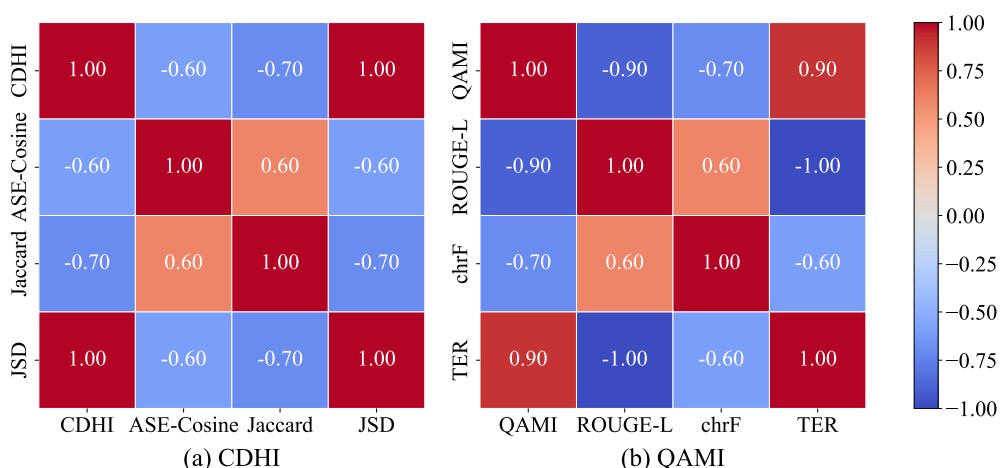

(a) CDHI  (b) QAMI

Figure 2: Spearman correlations between CDHI/QAMI and other metrics.

Based on the CNN/DailyMail dataset, we evaluate the validity of the proposed hallucination indexes on the untuned T5 family, Qwen2.5-Instruct, and Mistral-v0.1. Technically speaking, we evaluate the validity by examining how CDHI/QAMI correlate with other established metrics.

For CDHI, we compare it with contextual consistency metrics like Adjacent-Sentence Embedding Cosine (ASE-Cosine) (Gao et al., 2021; Reimers & Gurevych, 2019), Jaccard Overlap (Jaccard) (Jaccard, 1901; Shao et al., 2024), and topic Jensen–Shannon Divergence (JSD) (Lin, 2002; Wang et al., 2024). For QAMI, we compare it with classical summary accuracy metrics, such as Recall-Oriented Understudy for Gisting Evaluation-Longest Common Subsequence (ROUGE-L) (Lin, 2004; Saha & Zhang, 2023), character n-gram F-score (chrF)(Popović, 2015; Winata et al., 2024), and Translation Edit Rate (TER) (Snover et al., 2006; Deguchi et al., 2024). We take the first 100 samples from the validation set after shuffling it with a fixed random seed of 42 and conduct 10 trials. After averaging model-wise performance, we compute Spearman correlation coefficients between metrics mentioned above.(Spearman, 1961; Zhang & Jia, 2021).

As shown in Figure 2(a), CDHI has moderate negative correlations with ASE-Cosine and Jaccard ($-0.60$, $-0.70$) and a strong positive correlation with JSD (1.00). This suggests that higher CDHI indicates weaker contextual dependence and greater topical drift, effectively capturing the contextual disconnection hallucination.

Figure 2(b) demonstrates that QAMI has strong negative correlations with accuracy metrics ROUGE-L and chrF ($-0.90$, $-0.70$), and a strong positive correlation with TER (0.90). This indicates that QAMI effectively measures QA mismatch hallucination: smaller values show generations closely aligned with question semantics, leading to greater overlap with the reference summary and higher overlap-based scores.

From a practical perspective, it is crucial that both CDHI and QAMI are reference-free, while ROUGE-L, chrF, and TER are reference-dependent. In cases where references are unavailable, costly, or biased, the proposed metric suite provides a significant advantage.

|  | T5-small | T5-base | T5-large | Mistral-v0.1 | Qwen2.5-Instruct | Pearson $r$ | Spearman $\rho$ |
|---|---|---|---|---|---|---|---|
| **JSD** | 0.73 | 0.92 | 0.93 | 0.57 | 0.57 | 0.84 | 1.00 |
| **CDHI** | -13.83 | -13.21 | -12.68 | -27.06 | -19.66 | | |

Table 1: Correlation between CDHI and JSD across Models.

JSD uses Jensen–Shannon divergence to measure differences in topic distributions between texts, reflecting the topical consistency between generated text and original context (Chang et al., 2023; Calderon et al., 2024). Thus, we use JSD as the comparison metric for CDHI. Table 1 shows a clear positive correlation between CDHI and JSD: T5-large has the highest CDHI, corresponding to the largest JSD and indicating more contextual disconnection hallucination. Conversely, Mistral-v0.1 and Qwen2.5-Instruct show lower CDHI and the smallest JSD, suggesting better contextual alignment. Besides, CDHI and JSD have a Pearson correlation of 0.84 and a Spearman correlation of 1.00, confirming CDHI's effectiveness in quantifying the contextual disconnection hallucination.

|  | T5-small | T5-base | T5-large | Mistral-v0.1 | Qwen2.5-Instruct | Pearson $r$ | Spearman $\rho$ |
|---|---|---|---|---|---|---|---|
| **ROUGE-L** | 27.72 | 27.21 | 31.27 | 17.34 | 24.81 | -0.895 | -0.90 |
| **QAMI** | -502.13 | -594.38 | -1024.95 | -164.49 | -342.36 | | |

Table 2: Correlation between QAMI and ROUGE-L across Models.

ROUGE-L is the de facto benchmark for accuracy in summarization tasks. It focuses on the Longest Common Subsequence, emphasizing sentence structure and assessing lexical overlap in summary quality. We evaluate QAMI against ROUGE-L across models. As shown in Table 2, T5-large has the smallest QAMI and the highest ROUGE-L, indicating the least QA mismatch hallucination, while Mistral-v0.1 shows the largest QAMI. Besides, QAMI and ROUGE-L have a Pearson correlation of -0.895 and a Spearman correlation of -0.90, confirming QAMI's effectiveness in quantifying the QA mismatch hallucination.

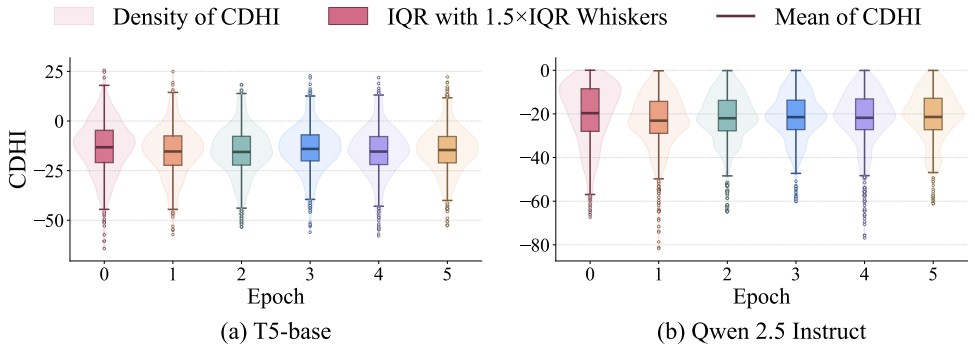

Figure 3: CDHI varying trajectories over the training process.

### 4.2 HALLUCINATION DYNAMIC TRACK

In a standard supervised training setup, we finetune T5-base and Qwen2.5-Instruct on the CNN/DailyMail dataset for five epochs, tracking hallucination dynamics during learning. After each epoch, we compute QAMI and CDHI on the validation split, plotting their empirical distributions and epoch-wise trajectories of summary statistics. These diagnostics help analyze the evolution of the contextual disconnection hallucination and the QA mismatch hallucination as training steps increase and generated sequence length grows.

**CDHI training dynamics.** Figure 3 illustrates the training trajectories of CDHI. During training, both models' CDHI distributions mostly remain below zero, indicating that generations remain

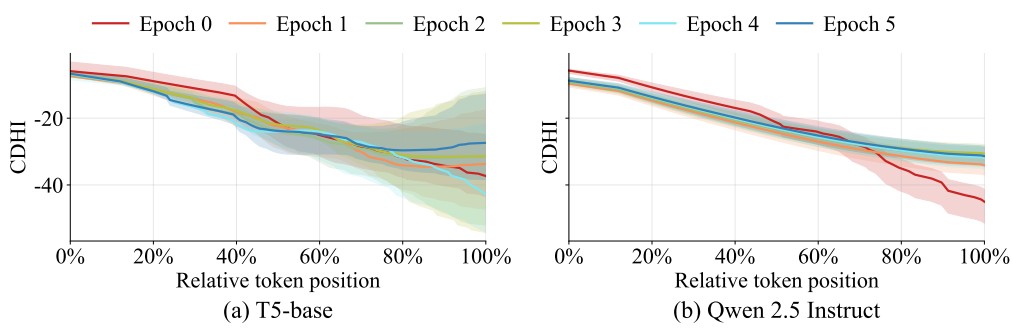

Figure 4: Per-epoch mean CDHI (±95% CI) vs. relative token position.

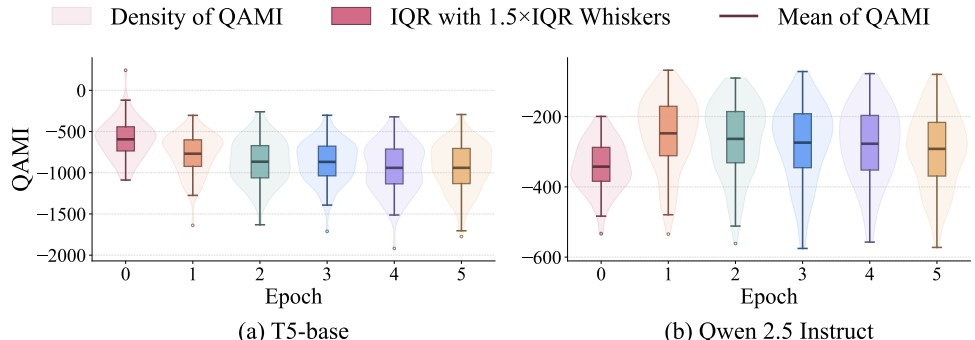

Figure 5: QAMI varying trajectories over the training process.

strongly constrained by prior context. From epoch 0 to epoch 1, the mean CDHI declines significantly, followed by minor fluctuations. Both models show distributional contraction: the Inter Quartile Range (IQR) and IQR whiskers shrink modestly, suggesting reduced variability and more consistent behavior.

**CDHI token position dynamics.** Figure 4 shows the per-epoch mean CDHI by relative token position, with the solid curve indicating the mean and the shaded band the 95% confidence interval (±95%CI). Relative token position normalizes each token's index within a sequence and expresses its location from 0% (beginning) to 100% (end). As position increases, CDHI decreases overall, indicating a diminishing propensity for the contextual disconnection hallucination as the sequence continues. This pattern is consistent with autoregressive intuition: early tokens must simultaneously establish topic and structure (we hypothesize this contributes to higher CDHI), whereas later tokens are more strongly conditioned by the generated prefix. Comparing architectures, CDHI declines for both models; however, T5-base exhibits a sharper late-position drop with wider end-of-sequence variance, while Qwen2.5-Instruct shows a steadier decline with tighter per-epoch clustering. Overall, newly generated tokens increasingly align with the preceding output as the sequence unfolds.

**QAMI training dynamics.** As shown in Figure 5, the QAMI distribution shifts toward more negative values across epochs during supervised training. For Qwen2.5-Instruct in Figure 5(b), QAMI rises sharply from epoch 0 (untrained) to epoch 1, then declines. For T5-base and Qwen2.5-Instruct alike, both models experience a faster decrease in QAMI early on, stabilizing later. This behavior indicates models rely more on the input query/instruction as training progresses, while the QA mismatch hallucination is suppressed. The temporary QAMI increase for Qwen2.5-Instruct at epoch 1 results from its rapid adaptation to generic summarization templates; T5-base establishes query-conditional dependence earlier, leading to a steady decline in QAMI from the start.

### 4.3 DISCUSSION

We introduce the proposed index system including CDHI and QAMI from a causal perspective that characterize hallucinations of the contextual disconnection and the QA mismatch, shifting evaluation from static scores to a traceable process diagnosis. CDHI and QAMI estimate the causal responsiveness of output to the question and prefix respectively, without relying on gold references.

They are suitable for settings with scarce or biased references, which demonstrate strong convergent validity with mainstream reference-based measures of consistency and accuracy.

Experiments show that most of hallucination risk decreases rapidly in early training, then enters a plateau or diverges across settings; as relative token position advances along the sequence, newly generated tokens increasingly align with preceding output, with the initial segment being the most vulnerable. These dynamics suggest concentrating monitoring and mitigation on early training checkpoints and the initial segment of generation, where targeted objectives and decoding controls are most effective at suppressing hallucinations.

Our experiments show that decoder-only models (e.g., Qwen2.5-Instruct, Mistral-v0.1) exhibit more severe QA mismatch hallucinations, initially increasing with fine-tuning but later declining; thus, early fine-tuning should focus on suppressing these hallucinations. Encoder–decoder models (e.g., T5 family) demonstrate a higher risk of contextual disconnection hallucinations that decrease as token position advances; therefore, stabilizing early context alignment should be prioritized.

## 5 RELATED WORK

LLM hallucinations of faithfulness can be generally categorized into two types: the *contextual disconnection hallucination* and the *QA mismatch hallucination*. The former involves internal consistency, resulting in logical contradictions or semantical breaks with the preceding sequence. The latter pertains to alignment with the user's input, where misinterpretation of intent causes the response to deviate from the question.

For contextual disconnection hallucinations, Reimers & Gurevych (2019) propose the Adjacent-Sentence Embedding cosine (ASE-Cosine) to evaluate sentence-level semantic continuity through vector similarity to identify local coherence breaks. Jaccard (1901) puts forward the Jaccard Overlap to measure the decline of shared lexical content between context and generation. Lin (2002) proposes the topic Jensen–Shannon Divergence, which assesses thematic distribution shifts between input and output to determine the degree of "sliding away" from the prior context. Together, these three signals—local semantics, lexical sharing, and topic-level distributions—effectively provide complementary evidence to diagnose contextual disconnection hallucinations. (Gao et al., 2021)(Shao et al., 2024)(Wang et al., 2024)

For the QA mismatch hallucination, the research community typically relies on reference-based alignment methods. For summarization and open-ended generation, ROUGE-L (Lin, 2004) measures the Recall-Oriented Understudy for Gisting Evaluation-Longest Common Subsequence. For example, QAGS proposed by Wang et al. (2020) and QAFactEval proposed by Fabbri et al. (2022) show changes in ROUGE-L alongside QA-based consistency scores, while SummaC proposed by Laban et al. (2022) uses ROUGE-L as a baseline to differentiate high surface overlap from lower faithfulness. Meanwhile, chrF (character n-gram F-score) and TER (Translation Edit Rate) quantify character-level overlap and minimum edit effort relative to references. Raunak et al. (2021) report chrF/TER in analyzing off-source/oscillatory hallucinations in NMT, while Deguchi et al. (2024) use chrF and TER as key indicators for a translation hallucination detector.

## 6 CONCLUSION

In this work, we introduce a *Causal Analytical Framework for Time-Varying Dynamics of Hallucinations* in LLMs. Building on Structural Causal Modeling, we derive deconfounded, time-varying index system that quantify the contextual disconnection and the QA mismatch hallucinations. Without gold references, our index system exhibits clear correlations with established reference-based metrics. Experiments show that the contextual disconnection hallucination decreases through training in general, then enters a plateau. Besides, the QA mismatch hallucination diverges across models. The finding enlightens us that architecture matters. Decoder-only models warrant early suppression of the QA mismatch, while encoder–decoder models benefit from stabilizing early context alignment. As token position advances in the sequence, generated tokens increasingly align with preceding output, with the initial segment being most vulnerable. To sum up, what we put forward can not only be used to evaluate LLM hallucinations without relying on standard references, but also help guide more research on reducing hallucinations in complex tasks.

## ETHICS STATEMENT

This work does not involve human subjects, animal experiments, or any potentially harmful insights. The datasets used in the experiments are publicly available, and all data processing steps were performed in compliance with the relevant privacy and security regulations. The authors confirm that they have read and adhered to the ICLR Code of Ethics. No conflicts of interest exist in this study, and no financial sponsorship influenced the research.

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

# A APPENDIX

## A.1 THE USE OF LARGE LANGUAGE MODELS (LLMS)

In this work, Large Language Models (LLMs) are only employed to assist with language polishing and writing refinement. The LLM did not influence content ideation, data analysis, or experimental design in any way.

