# OpenReview forum: "Causal Analytical Framework for Time-Varying Dynamics of Hallucinations in LLMs"
_ICLR.cc/2026/Conference — ICLR 2026 Conference Withdrawn Submission_

### Official Review · Reviewer_gTuf · 2025-10-29

**Soundness:** 2
**Presentation:** 3
**Contribution:** 2
**Rating:** 4
**Confidence:** 2

**Summary:**

This paper proposes a causality-based hallucination detection framework that employs SCMs to quantitatively characterize two types of hallucinations in LLMs from a temporal evolution perspective: contextual disconnection and question–answer mismatch. The authors design two computable metrics—Contextual Disconnection Hallucination Index and Question–Answer Mismatch Index—and validate their effectiveness and dynamic variation across several mainstream models.

**Strengths:**

1. This work is the first to introduce a SCM to explain the dynamic causal chain underlying hallucination generation, demonstrating a clear degree of novelty.
2. The mathematical derivations are well-structured and transparent, providing a solid theoretical foundation for subsequent studies.
3. The proposed metrics show high correlations with established measures such as ROUGE and JSD, thereby validating the theoretical consistency of the framework.

**Weaknesses:**

1. The paper does not demonstrate the performance of the proposed metrics on real-world hallucination detection tasks.
2. It lacks algorithmic details on how to efficiently estimate $P(A_t \mid do(A_{t-1}))$ within LLMs.
3. The evaluation is limited to summarization tasks, which is insufficient to support the claim of a "general hallucination detection framework."
4. The related work section fails to cover the latest advances in hallucination detection from 2024–2025.
5. The paper does not clearly distinguish between causal interpretability analysis and hallucination detection evaluation.

**Questions:**

1. Please clarify how $P(\epsilon_t \mid Q, A_{t-1})$ is estimated in practice—e.g., via approximations using model logits or through Monte Carlo sampling.
2. Have the metric’s stability and robustness been validated on larger-scale datasets?
3. If the input question is ambiguous, can the causal effect captured by QAMI still disentangle semantic shift from hallucination?
4. Could you present an end-to-end causal intervention experiment to demonstrate the controllability of CDHI/QAMI?
5. Do you plan to leverage these metrics for hallucination mitigation in future work? For example, decoding contro.

---

### Official Review · Reviewer_zGqE · 2025-11-02

**Soundness:** 2
**Presentation:** 2
**Contribution:** 2
**Rating:** 4
**Confidence:** 3

**Summary:**

This paper introduces a causal modeling approach to study hallucinations in Large Language Models (LLMs), proposing two new metrics — the Contextual Disconnection Hallucination Index (CDHI) and the Question-Answer Mismatch Hallucination Index (QAMI). The authors build upon the Structural Causal Model (SCM) framework to isolate causal from correlational dependencies in LLM output generation, aiming to quantify hallucination dynamics without reliance on ground-truth references. Experiments are performed on summarization datasets (mainly CNN/DailyMail) using several models, including T5 variants, Mistral, and Qwen2.5, to validate the correlation of these proposed metrics with standard reference-based measures such as ROUGE-L, chrF, and JSD.

**Strengths:**

Strengths

The paper tackles an important and timely issue — hallucinations in LLMs — from a novel angle using causal inference tools.

The integration of Pearlian causal reasoning into LLM analysis is conceptually interesting and potentially opens a new research direction.

The proposal to move toward reference-free hallucination detection aligns well with the broader challenge of evaluating generative models without gold-standard data.

The authors provide mathematical derivations and clearly state their theoretical assumptions, maintaining technical rigor throughout.

The experiments include multiple models, lending at least superficial generality to the findings.

**Weaknesses:**

Weaknesses

Lack of originality in causal modeling: The causal formulation (using SCMs and backdoor adjustment) is textbook-level and largely rephrases standard causal inference machinery without offering any substantive innovation tailored to LLM generation dynamics. The “causal graph” used is trivial and self-evident in any autoregressive model, offering little explanatory depth beyond existing token-dependency structures.

Questionable empirical rigor: The experimental validation is weak and largely correlational, ironically undermining the paper’s own emphasis on causality. The analysis only measures correlations between proposed indices and known metrics, which does not constitute a causal or even independent validation. There are no ablation studies, robustness checks, or counterfactual simulations to support causal claims.

Overstated theoretical claims: The manuscript repeatedly claims to achieve “unbiased causal estimation” and “true causal effect isolation,” but in practice the entire procedure relies on model-generated probabilities rather than real interventions. No actual causal interventions (such as controlled token perturbations or prefix replacements) are performed, meaning the results remain correlational by construction.

Poor connection to hallucination behavior: The proposed indices (CDHI, QAMI) appear mathematically contrived and lack intuitive interpretability. There is no qualitative analysis or human evaluation to confirm whether higher CDHI/QAMI actually corresponds to more hallucination-prone outputs.

Dataset and task mismatch: Using summarization datasets like CNN/DailyMail to study hallucinations in instruction-tuned LLMs is conceptually limited. The framework may not generalize to open-domain generation, question answering, or reasoning tasks where hallucinations are most problematic.

Presentation issues: The writing is verbose and at times tautological, frequently repeating the same conceptual claims without adding clarity. Figures (e.g., correlation heatmaps) are simplistic and do not substantiate causal findings.

**Questions:**

How do the authors justify calling their method “causal” when no actual interventions are performed and all quantities are derived from model likelihoods?

What evidence supports that CDHI and QAMI genuinely measure hallucinations rather than general coherence or fluency?

How do the proposed metrics behave in tasks beyond summarization — for example, factual question answering or multi-turn dialogue — where hallucination manifests differently?

Could the observed correlations with ROUGE and JSD simply reflect general lexical or topical overlap rather than any causal insight?

How sensitive are CDHI and QAMI to model size, sampling temperature, or dataset domain — are these indices stable across different conditions?

The framework presumes access to token-level probabilities. How feasible is this for closed-source or API-based models?

---

### Official Review · Reviewer_25xN · 2025-11-02

**Soundness:** 2
**Presentation:** 2
**Contribution:** 2
**Rating:** 2
**Confidence:** 4

**Summary:**

From a causal inference perspective, this paper proposes two metrics, CDHI and QAMI, to quantify the consistency of language model outputs. Experiments are conducted to demonstrate the rationality of these metrics.

**Strengths:**

The paper introduces a causal-inference-based perspective to investigate contextual consistency in language models, offering a novel and potentially valuable angle for this research problem.

**Weaknesses:**

1.	Lack of justification for the proposed causal graph. The paper does not provide sufficient explanation for the rationality of its proposed causal graph. In my view, the presented graph is more suitable for encoder–decoder architectures, whereas the causal chain $Q \rightarrow A_1 \rightarrow \dots \rightarrow A_t$ better aligns with decoder-only language models commonly used today.
2.	Questionable feasibility of CDHI computation. The proposed CDHI metric involves computing a full expectation over the question Q. It is unclear how this computation is feasible given the enormous question space. The authors need to elaborate on the computational tractability and complexity of this process to convince readers of its practicality.
3.	Inconsistent use of terminology related to hallucination. The authors use contextual consistency metrics to assess output consistency and summary accuracy metrics to measure question–answer relevance. However, these definitions diverge considerably from mainstream hallucination detection research, which focuses on identifying and localizing misalignment with real-world facts. The authors should reconsider their problem statements and avoid using the term ``hallucination’’, which may mislead readers.
4.	Limited and outdated experimental setup. All experiments are conducted solely on the CNN/DailyMail dataset, which significantly limits the generality of the conclusions. The models use, T5, Mistral-v0.1, are relatively outdated, and their behavior may differ from modern large language models (e.g., Llama-4). Moreover, the authors fail to specify the size of the Qwen2.5 model, leaving readers uncertain. A statement on reproducibility and publicly available code would greatly enhance the paper’s credibility.

**Questions:**

1.	In Figure 2, why are all Spearman correlation values around the order of 1e-1? Such uniform precision looks suspiciously consistent.
2.	As mentioned in Weakness, how is CDHI computed in practice during deployment?

---

### Note · Authors · 2025-12-03

I have read and agree with the venue's withdrawal policy on behalf of myself and my co-authors.